# Indirect Assessment of Railway Infrastructure Anomalies Based on Passenger Comfort Criteria

Patricia Silva [1,2,*] , Diogo Ribeiro [3] , Pedro Pratas [4] , Joaquim Mendes [4,5] and Eurico Seabra [2]

1  CONSTRUCT-LESE, Faculty of Engineering, University of Porto, 4099-002 Porto, Portugal
2  Department of Mechanical Engineering, School of Engineering, University of Minho,
   4710-057 Guimarães, Portugal; eseabra@dem.uminho.pt
3  CONSTRUCT-LESE, School of Engineering, Polytechnic of Porto, 4249-015 Porto, Portugal; drr@isep.ipp.pt
4  INEGI, LAETA, 4099-002 Porto, Portugal; ppratas@inegi.up.pt
5  Faculty of Engineering, University of Porto, 4099-002 Porto, Portugal
*  Correspondence: ppsilva@fe.up.pt

**Abstract:** Railways are among the most efficient and widely used mass transportation systems for mid-range distances. To enhance the attractiveness of this type of transport, it is necessary to improve the level of comfort, which is much influenced by the vibration derived from the train motion and wheel-track interaction; thus, railway track infrastructure conditions and maintenance are a major concern. Based on discomfort levels, a methodology capable of detecting railway track infrastructure failures is proposed. During regular passenger service, acceleration and GPS measurements were taken on Alfa Pendular and Intercity trains between Porto (Campanhã) and Lisbon (Oriente) stations. ISO 2631 methodology was used to calculate instantaneous floor discomfort levels. By matching the results for both trains, using GPS coordinates, 12 track section locations were found to require preventive maintenance actions. The methodology was validated by comparing these results with those obtained by the EM 120 track inspection vehicle, for which similar locations were found. The developed system is a complementary condition-based maintenance tool that presents the advantage of being low-cost while not disturbing regular train operations.

**Keywords:** preventive maintenance; railways; passenger comfort; vibration; ISO 2631

## 1. Introduction

Railways are one of the most widely used public transportation systems, mainly because of their high transportation capacity, reduced boarding time, and the possibility of making better use of the journey time to work or enjoy train facilities. Additionally, due to their low environmental impact, multiple governments are promoting their use as a mass transportation mode. Following this recommendation, train passengers have continuously increased from 2013 until 2020 (the year of the COVID-19 pandemic). Between 2015 and 2019, an increase of 10% was found in Europe, and approximately 4000 billion passenger kilometres were recorded worldwide. During the COVID-19 pandemic, train passenger numbers decreased to historical levels due to multiple lockdowns [1–3].

Currently, the number of passengers is even higher than in the pre-pandemic period [4]. Increasing trains' attractivity and journey comfort will ensure the continuation of this trend. Passengers define a comfortable ride based on safety, comfort, and user conditions [5,6]. High safety levels are guaranteed through adequate maintenance actions based on corrective or preventive strategies. The corrective action occurs after a fault is recognised and intends to recover the normal working state [7–10]; in contrast, proactive measures intend to prevent and minimise eventual faults that did not yet cause the system to fail, thus anticipating the problems.

The main goal of preventive maintenance is to preserve system functions and prevent rail system failure. It is performed according to a defined scheduled time. On the other

hand, condition-based maintenance (CBM), the typical railway track infrastructure condition monitoring system, is a preventive maintenance strategy that acts when there is evidence of failure. Railway track infrastructure monitoring actions can be performed in two ways, either by human or by automatic means, depending on the type and extent of the work required [11–13]. Human inspection is performed by well-trained inspectors who periodically walk along railway track infrastructure to detect defects. The inspectors can be aided by portable equipment for the auscultation and measurement of the geometric parameters of the track infrastructure. However, this maintenance technique results in high costs and can be potentially hazardous for inspectors. Moreover, human inspection can only be applied by stopping or restraining traffic, and its results highly depend on the observer's capability of detecting anomalies and recognising critical situations [7,14]. Due to the problems associated with human inspection and advances in technology, automatic inspection methods have been developed, such as using inspection vehicles. These dedicated vehicles can detect railway track infrastructure defects and evaluate infrastructure performance. Generally, an inspection vehicle uses optical and inertial sensors linked to a global positioning system (GPS), which increases the method's efficiency and decreases the required time [8,9,15,16]. This inspection is periodically performed along the railway track infrastructure. However, in addition to being expensive, these vehicles introduce traffic disruptions during the inspection, affecting regular service operations. Different types of inspection vehicles are used worldwide. In Portugal, the EM 120 inspection vehicle, owned by Infraestruturas de Portugal (IP), is responsible for identifying maintenance needs and railway track infrastructure failures. After detecting a failure, the defect is fixed and, ideally, the system is restored to its initial state [10,12]. More recently, cargo vehicles have been used as instrumented inspection vehicles, particularly in the axle box [17–22]. However, implementing the experimental setup is problematic for this approach, limiting its use in passenger trains.

These vehicles have the advantage of continuously surveying the track without any traffic interference, thus providing information on a more regular basis at a reduced cost. Moreover, railway track infrastructure maintenance also improves the interaction performance of the rail vehicle and the overhead infrastructures, such as the pantograph-catenary interaction, which leads to a more safe, stable, and comfortable journey [23].

It is known that there is a significant difference in vibration levels when comparing a healthy railway track infrastructure with a defective one, which is characterised by higher peak values. Thus, besides affecting safety, isolated and continuous infrastructure defects lead to poor passenger comfort due to increased vibration levels [9]. Once comfort is significantly affected by vibration, which is mainly caused by train motion and rail track infrastructure irregularities, passengers are subjected to discomfort throughout their journey due to contact with the seat and floor. Therefore, vibration is transmitted through all passenger-seat and passenger-floor contact surfaces; this is generally defined as whole-body vibration (WBV). Besides causing discomfort, this can also lead to fatigue and, in some extreme cases, diseases. Due to the adverse consequences of WBV, it is vital to classify vibration levels in the rail environment. One of the international reference documents is the ISO 2631 standard, which precisely quantifies WBV regarding comfort, human health, and motion sickness [24–27].

Up to this date, passenger comfort levels have not been used for possible damage detection. Based on the close connection between railway track infrastructure conditions, induced vibrations, and passengers' discomfort levels, it was possible to develop a new CBM methodology to identify critical railway track infrastructure locations. The goal was to provide a low-cost solution to improve maintenance needs detection and increase infrastructure availability. The present method overcomes the limitations of traditional railway infrastructure maintenance detection methods as it does not cause a disturbance to railway operations. This study considers the railway track infrastructure to require maintenance actions if multiple trains with different dynamic characteristics present floor discomfort at the same GPS location.

Based on the limitations of previous experiments, this study aims to offer clear contributions regarding some aspects that, presently, according to the authors' knowledge, are not sufficiently addressed in the existing literature, particularly the following:

- Development of a methodology capable of detecting rail track infrastructure abnormalities or damages based on measurements on in-service trains. This way, the service runs under regular operation without disruption or interference.
- Development of a methodology that is easy to adapt and implement and can be applied in any in-service rail vehicle. This way, the limitations of the accelerometer installation in the axle box, which imposes difficulties on the processes of installation, maintenance, and dismounting, can be overcome.

The results of the present methodology are not dependent on the vehicle type. Thus, it can be applied to passenger trains with different characteristics. This was stated based on the results of multiple measurement campaigns proving the methodology's accuracy and precision.

## 2. Ballasted Track

Worldwide, as well as in Portugal, the most common railway system is the ballasted track system. Ballasted track system has lower construction costs and adequate responses to static and dynamic forces [28]. This track type is grouped into two main categories: superstructure and substructure. The former consists of rails, sleepers, and ballasts and components that connect these elements. In contrast, the latter is associated with the geotechnical system and comprises sub-ballasts, embankments, and subgrade layers [20,21,29,30]. Figure 1 presents a schematic representation of the railway ballasted track system.

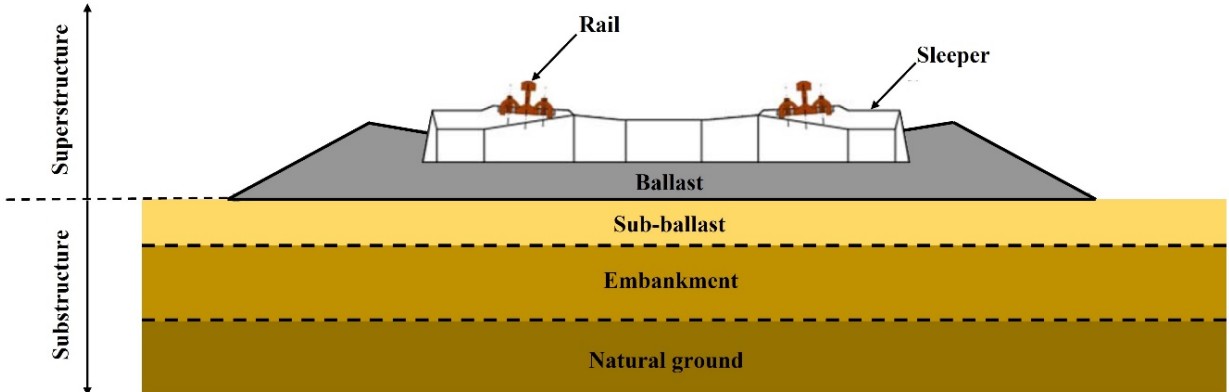

**Figure 1.** Schematic illustration of railway ballasted infrastructure.

Track integrity problems are dependent on the superstructure and substructure deformation. Both are related, but the health of the track substructure defines the structural performance [21,30].

The main substructure issues are related to sub-ballast and subgrade deterioration, which can induce poor drainage and track settlements. Those issues may also reduce track stability, capacity, and safety, leading to misalignments and increased wear [21].

Superstructure track faults can be grouped into three main conditions: those related to geometry (cross-level, alignment, longitudinal levelling, twist, and gauge), those dependent on rail surface faults (surface, corrugation, fatigue cracking, squat, and creep), and, lastly, those dependent on the ballast conditions (fouled ballast, ballast pockets, and poor ballast drainage) [20,28].

Track defects and irregularities negatively affect track performance and safety. Wavelengths between 10–120 m mainly influence passengers' comfort. EN 13848 [31] defines three wavelength intervals for the evaluation of vertical and lateral track geometry conditions: D1 (3–25 m), D2 (25–70 m), and D3 (70–150 m). The D1 wavelength range ir-

regularities are mainly associated with running safety conditions, whereas D2 and D3 ranges strongly affect ride comfort [31,32]. Lower wavelengths define rail surface faults; for example, rail corrugation can appear between 0.03–0.08 m of wavelength, whereas rail squat comprehends 0.02–0.04 m of wavelength. A discontinuity in the rail caused by a weld or joint can present 0.01–0.02 m of wavelength [33,34]. It should be highlighted that the aforementioned defects' wavelengths are the most typical ones. Nevertheless, depending on its severity and dimension, the same type of defect can produce higher or lower wavelengths. Figure 2 presents the rail track geometry conditions evaluation defined by EN 13848, along with rail surface faults (rail corrugation, squats, and discontinuities) and passenger comfort wavelengths.

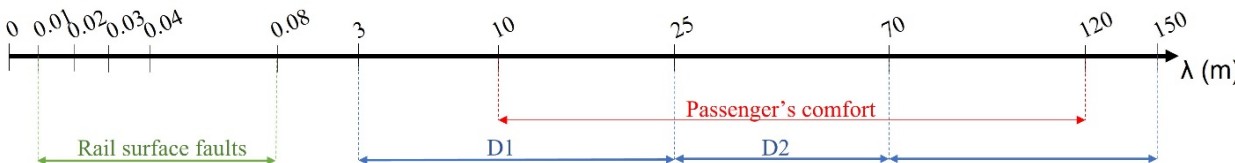

**Figure 2.** Typical rail surface faults and EN 13848 track geometry evaluation wavelengths and those capable of affecting passengers' comfort.

Rail defects and increased speed further amplify the rail vehicle–track dynamic interaction forces [19,20,30,35]. When a rail vehicle and track do not present defects or abnormalities, the vehicle exerts low-frequency forces (under 20 Hz). However, that frequency increases in the presence of faults, leading to dynamic impact wheel–rail forces. Moreover, dynamic loads, which are connected to track irregularities, amplify the rail deterioration rate. Vehicle vibration magnitudes are strongly linked with track irregularities and can be used to assess general track conditions effectively [19,29,30,36]. Karoumi et al. [37] and Norris [38] noticed that railway track defects could be identified based on acceleration records according to their peak values, as illustrated in Figure 3.

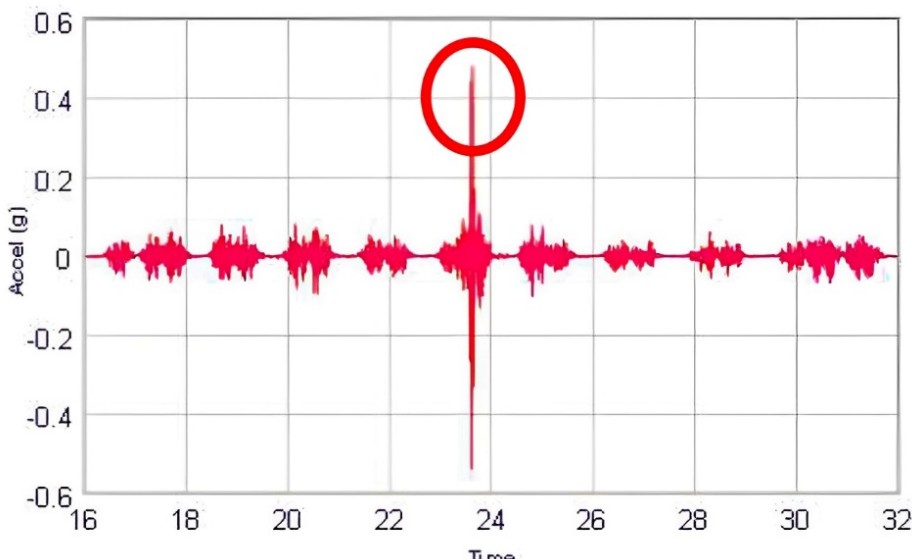

**Figure 3.** Detection of rail irregularities based on acceleration measurements (adapted from [35]).

Besides affecting comfort, track irregularities can lead to derailment, which can have significant consequences [18]. Therefore, promoting scheduled railway infrastructure maintenance is crucial to maintain high safety levels and the comfort of passengers.

When a railway vehicle passes over a railway track, it causes static (weight) and dynamic (inertia and impact) loads. That dynamic load leads the railway track to vibrate for a certain period and, consequently, deform. Moreover, those loads must be transmitted

to the track subgrade. The dynamic load exerted on the track is the critical parameter that causes track deterioration, whereas wear and fatigue are the main consequences. Generally, rail defects are caused by loads applied to the rails in longitudinal, transverse, and vertical directions. Additionally, the weather also plays a significant role in railway infrastructure deterioration as it can promote corrosion, thermal expansion, and buckling. This way, railway tracks experience different conditions at different points along the track and, consequently, different track sections show different degradation behaviours and require different maintenance plans [21].

## 3. Suspension Systems

The interaction between the wheels and the railway track infrastructure, be it healthy or with defects, generates vibrations with varying amplitude and frequency. These vibrations can cause damage to both trains and railway track infrastructures; thus, they must be controlled. The train suspension system, constituted of primary and secondary suspensions, fulfils this function by enabling the filtering of the vibrations derived from the vehicle–track dynamic interaction (promoting ride comfort) and controlling the kinematic modes of the bogie (promoting stability). Suspension systems are natural low-pass filters that prevent the transmission of high frequencies from the rail track and wheel to the carbody and seats [39].

The primary suspension's main goals are to secure stability and guidance, reduce track forces and wear, and improve curve performance. Regarding the secondary suspensions, it suppresses the vibration transmission from the bogie to the carbody and attenuates vehicle vibrations derived from railway track irregularities, improving ride comfort and controlling quasi-static motion [39–41].

Although with the same goal, different trains have different suspension systems, which interfere with the filtered ranges of frequencies. The present research conducted experimental tests on Alfa Pendular (AP) and Intercity (IC) trains. Due to their model series and tilting mechanisms, these trains present different suspension systems. Therefore, characterising those trains' suspension systems is essential.

### 3.1. Alfa Pendular Vehicle Suspension

The Portuguese AP tilting 4000 series train started its operation in 1999. This electric train has a total length of 158.9 m and reaches a maximum speed of 220 km/h. It is operated as a single unit comprising six cars, four motor units, and two trailers.

The AP train has an active tilting system, which reduces the lateral acceleration perceived by passengers and, consequently, allows for the performance of curves at higher speeds than the balanced one while maintaining high passenger comfort levels [39]. Two bogies support each car, distanced by 19 m, and are 6 m from the bogies of consecutive cars.

Figure 4a illustrates the primary AP suspension, while Figure 4b shows its secondary suspension system. The primary suspension is composed of four helicoidal springs (sets of two plus two) (1) and a vertical damper (2), both filtering the vibrations derived from the vehicle–track interaction in each wheel. Thus, each bogie of the AP tilting train has twelve flexi-coil springs combined with six dampers acting as a secondary suspension and sixteen springs coupled to four vertical hydraulic dampers performing as a primary suspension [42–44].

The mechanical elements of the secondary suspension ensure the carbody–bogie connection. The secondary suspension (see Figure 4b) is constituted of twelve flexi-coil springs grouped within four units of three (3). Between spring units, on each side of the bogie, there is a vertical (4), a transversal (5), and an anti-yaw (6) hydraulic damper.

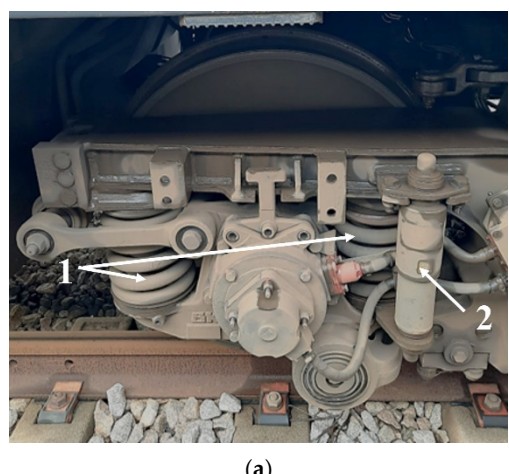

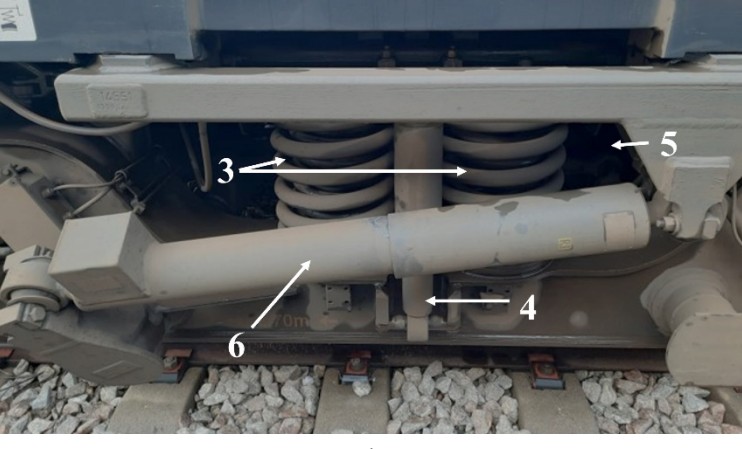

(**a**)  (**b**)

**Figure 4.** AP train bogie: (**a**) primary suspension; (**b**) secondary suspension.

*3.2. Intercity Vehicles Suspension*

The IC train service, introduced in 1980 in Portugal, is currently run by 5600 series locomotives with hauled Corail coaches [45]. These coaches were renovated in 2002 and can achieve a maximum speed of 200 km/h. The electric locomotive trails five Corail coaches, each with a length of 26.4 m. As on the AP train, each carriage has two bogies, separated by an 18.4 m distance within the same vehicle and an 8 m distance between consecutive carriages.

Figure 5 presents both primary and secondary suspension systems of IC vehicles [43]. The primary suspension (see Figure 5a) comprises eight helicoidal springs and four hydraulic dampers, resulting in two springs (1) and one damper (2) per wheel. The bogie–carbody connection, partially performed by the secondary suspension (Figure 5b), comprises two grouped helicoidal springs (3) aided by two vertical (4) and two transversal (5) dampers, one on each side of the bogie.

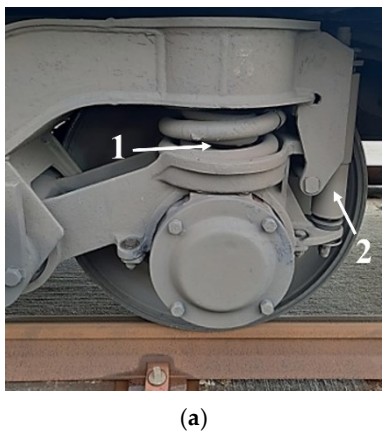

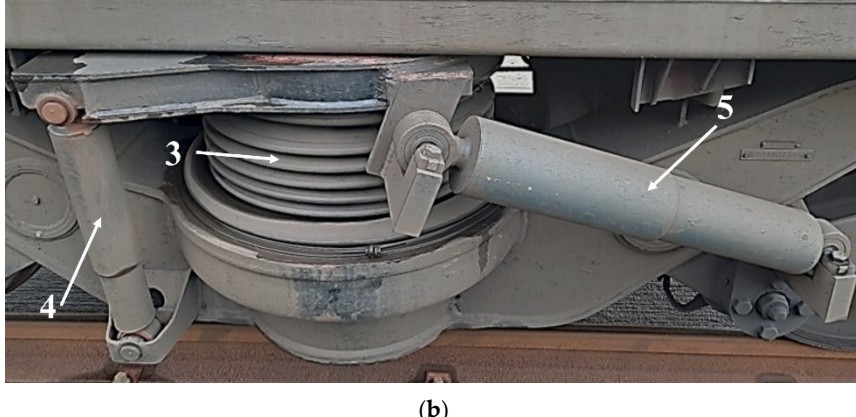

(**a**)  (**b**)

**Figure 5.** IC bogie: (**a**) primary suspension; (**b**) secondary suspension.

Table 1 shows the primary and secondary elements of both types of trains (AP and IC).

**Table 1.** Primary and secondary elements of AP and IC train suspension systems.

| Train | Primary Suspension | Secondary Suspension |
|---|---|---|
| AP | 16 helicoidal springs<br>+<br>4 vertical hydraulic dampers | 12 flexi-coil springs<br>+<br>6 hydraulic dampers<br>(2 vertical, 2 transversals, 2 anti-yaw) |
| IC | 8 helicoidal springs<br>+<br>4 vertical hydraulic dampers | 4 helicoidal springs<br>+<br>4 hydraulic dampers<br>(2 vertical, 2 transversal) |

## 4. Indirect Method for Infrastructure Condition Assessment Based on Comfort Criteria

Railway track infrastructure maintenance interventions are commonly decided based on measurements obtained by inspection vehicles and not on the dynamic response of in-service railway vehicles.

Passenger comfort is affected by the vibrations that the carbody experiences due to motion and railway track infrastructure irregularities. As discussed, railway track irregularities can be identified based on acceleration records according to visible peaks in those measurements. Passenger comfort levels are also assessed through acceleration measurements. Thus, a strong link between isolated railway track irregularities and passenger comfort levels can be identified. The present method relies on that connection. Based on the natural excitation created by the passage of railway vehicles over the track and on the fact that a defective railway track system induces higher vibration, it is expected that high discomfort levels will be accomplished when in the presence of an abnormality on the railway track infrastructure. Thus, it was hypothesised that railway track infrastructures require maintenance if multiple trains with different suspension mechanisms report instantaneous floor discomfort levels at the same geographic location.

Therefore, to investigate the defined hypothesis, a condition-based maintenance identification methodology was developed. The proposed methodology includes two stages: the experimental data measurement and acquisition and the data analysis. Acceleration measurements were performed based on a 3-axial accelerometer, aided by a precision GPS to determine the vehicle's location. The ISO 2631 standard provided the reference evaluation method for assessing passenger comfort. Then, a MATLAB algorithm was developed to match multiple railway vehicles' discomfort locations and identify maintenance needs locations. It should be highlighted that the methodology was applied on ballasted tracks, that is, the track type of the Northern line of the Portuguese railways. Nevertheless, there are no expected assessment differences when applying the present method to ballastless tracks.

### 4.1. Whole-Body Vibration Evaluation

The ISO 2631 standard quantifies WBV concerning comfort, health, and motion sickness. Frequencies between 0.5 and 80 Hz affect the body as a whole. Thus, these frequencies are defined as the most relevant ones. The standard defines 3-axial measurements on the vibration transmission interfaces: floor, seat surface, and seatback. The root-mean-square (rms) acceleration is calculated for each axis from the measurements. Depending on the human body characteristics, vibrations with similar intensities but different spectral content will induce different dynamic responses. To quantify this effect, the standard determines the application of weighting curves, assigning different weights to the rms acceleration and rating its impact on the human body [46–49]. The weighting process is calculated according to Equation (1):

$$a_w = \left[ \sum (W_i a_i)^2 \right]^{\frac{1}{2}} \tag{1}$$

where $W_i$ represents the weighting frequencies and $a_i$ represents the rms accelerations. Weighting curve application relies on the measurement location and purpose. This way, the effect of the frequencies most influencing passengers' discomfort is amplified, while

that of the frequencies with less of an impact on discomfort is reduced. The total vibration ($a_v$) is obtained following Equation (2):

$$a_v = \left( k_x^2 a_{wx}^2 + k_y^2 a_{wy}^2 + k_z^2 a_{wz}^2 \right)^{\frac{1}{2}}$$ (2)

where $a_w$ is the rms accelerations for each axis and $k$ represents the multiplying factor dependent on the measuring position according to Table 2.

**Table 2.** Frequency weighting curves and multiplying factors defined by ISO 2631 for comfort analysis of a seated passenger.

|  | X-Axis | Y-Axis | Z-Axis |
|---|---|---|---|
| Floor | $W_k$ and $k_x = 0.25$ | $W_k$ and $k_y = 0.25$ | $W_k$ and $k_z = 0.40$ |
| Seat surface | $W_d$ and $k_x = 1.0$ | $W_d$ and $k_y = 1.0$ | $W_k$ and $k_z = 1.0$ |
| Seatback | $W_c$ and $k_x = 0.80$ | $W_d$ and $k_y = 0.50$ | $W_d$ and $k_z = 0.40$ |

Lastly, based on $a_v$, a defined scale evaluates discomfort levels (see Table 3), in which values higher than 0.315 m/s$^2$ are considered uncomfortable.

**Table 3.** ISO 2631 comfort evaluation scale. Adapted from [47].

| $a_v$ (m/s$^2$) | Ride Comfort |
|---|---|
| $\leq$0.315 | Not uncomfortable |
| 0.5–0.63 | A little uncomfortable |
| 0.63–0.8 | A little uncomfortable to fairly uncomfortable |
| 0.8–1.0 | Fairly uncomfortable to uncomfortable |
| 1.0–1.25 | Uncomfortable |
| 1.25–1.6 | Uncomfortable to very uncomfortable |
| 1.6–2.0 | Very uncomfortable |
| 2.0–2.5 | Very uncomfortable to extremely uncomfortable |
| $\geq$2.5 | Extremely uncomfortable |

As mentioned, the standard determines acceleration measurements at interface surfaces where vibrations are transmitted to the user. However, it is well known that the seat modifies vibration transmission [50–53]. Additionally, the floor represents the closer location between the users and the carbody. This way, the present research applies the calculation of instantaneous floor discomfort by performing the standard analysis per second and applying the recommendations concerning floor measurement location.

### 4.2. Condition-Based Maintenance Identification Methodology

The developed condition-based maintenance identification methodology comprises two stages: data acquisition and data analysis. The latter, performed in MATLAB, produces a list of locations where high vibration levels were found in all railway vehicles, thus identifying the exact track locations that possibly require maintenance.

#### 4.2.1. Data Acquisition System

The data acquisition system comprises one 3-axial accelerometer and a GPS measurement; both components log data to a μSD card. Sensors were mounted on board multiple trains. Their location inside the rail train was defined based on points corresponding to the train's beginning, middle, and end. Moreover, each of these locations was defined as to the rear of the respective bogie.

Floor vibration measurements were conducted by the 3-axial accelerometer (PCE-VDL-24I, PCE Instruments, Southampton, UK), with a measurement range of $\pm 16$ g, resolution of 0.004 g, and sampling rate between 0 and 2400 Hz, which permits data recording and follows ISO 2631 requirements [54]. As discussed, the standard defines the frequency range between 0.5 and 80 Hz as the most relevant for comfort evaluation. Thus, a sample rate of 200 Hz was set, respecting the Nyquist theorem and preventing aliasing.

The GPS geographic location was obtained from a system composed of a GPS logger [55] connected with a RedBoard Qwiic [56]; it was programmed to retrieve data concerning location and rail vehicle speed and record them on a µSD card at 1 Hz. For future data processing analysis, it is fundamental to synchronise data; thus, vibration and geographic measurements were obtained synchronously. Figure 6 shows the data acquisition system, including the vibration measurement and GPS units.

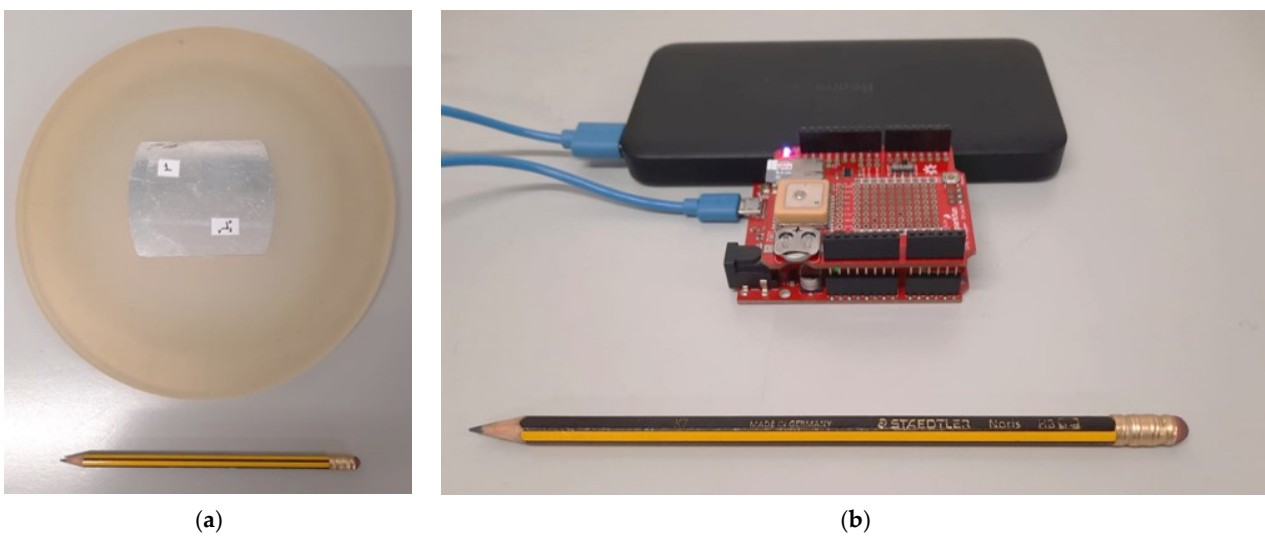

(**a**)    (**b**)

**Figure 6.** Data acquisition system: (**a**) 3-axial accelerometer pad; (**b**) GPS system.

As discussed, experiments were run on multiple AP and IC trains, which have different structural designs. Figure 7 illustrates both types of trains.

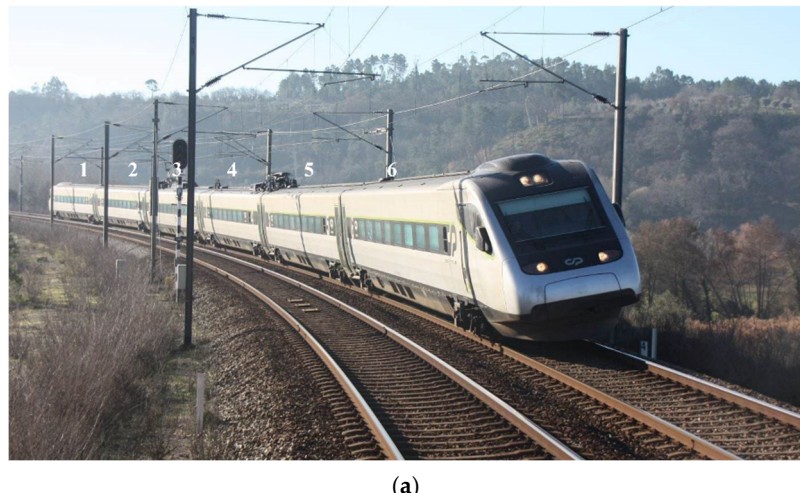

(**a**)

**Figure 7.** *Cont.*

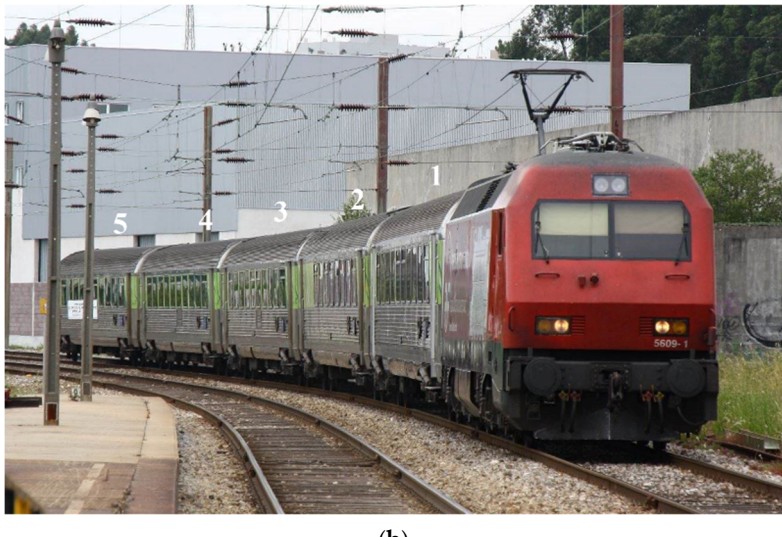

**(b)**

**Figure 7.** Experimented vehicles: (**a**) AP train; (**b**) IC train.

4.2.2. Data Analysis

The data analysis was performed in MATLAB 2022a [57] using an algorithm developed for this purpose. The algorithm starts by calculating instantaneous floor discomfort (discomfort level at each second) according to ISO 2631 recommendations for each rail journey. A slight adaptation of the standard was conducted as discomfort levels were divided into only two categories instead of the nine defined for the standard. Those categories were defined based on the first discomfort threshold of the ISO 2631 standard. Table 3 presents all comfort levels defined according to the standard. The first category determines that accelerations equal to or under 0.315 m/s² are ranked as "Not uncomfortable", whereas, above 0.315 m/s², accelerations are rated as "Uncomfortable". The uncomfortable locations were identified and mapped in all journeys.

To remove the interference of eventual railway vehicle maintenance needs, it was assumed that only the common discomfort sections required maintenance, i.e., segments with the same location and simultaneously reported by different trains.

The GPS signal was perfectly obtained at any location inside the IC train. This way, geographic location and acceleration records were acquired at the same location. However, the same conditions were not present in AP vehicles, where the best GPS signal was obtained by installing the device in the train driver's cabin. Therefore, considering the AP train length, a maximum distance of 158.9 m between the accelerometer and GPS model could be observed for measurements taken at the opposite end of the train. Moreover, considering the worst-case scenario where the AP travels at 220 km/h, the distance it achieves in 1 s is approximately 61.1 m. Thus, a maximum offset of 220 m (158.9 + 61.1 m) between AP and IC was used to obtain the matching segment.

Figure 8 illustrates the steps between taking the measurements and producing the final report.

It should be highlighted that the developed methodology only identifies abnormalities in the railway track infrastructure and, therefore, does not distinguish the type of abnormality presented. Moreover, the developed methodology is more accurate for identifying abnormalities within 10–120 m of wavelength once it is based on comfort analysis. Lower wavelength irregularities may not be precisely detected.

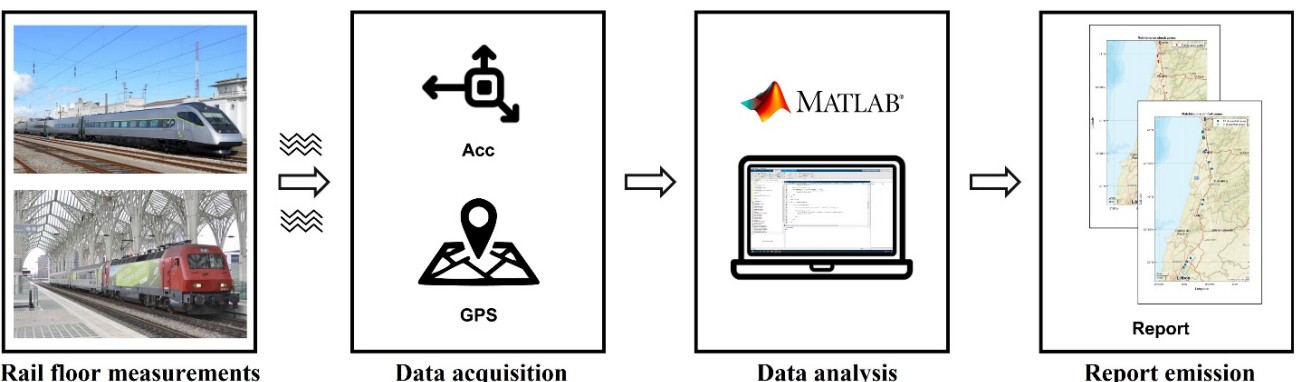

**Figure 8.** Developed methodology illustration.

## 5. Case Study: Portuguese Northern Line Condition Assessment

To validate and verify the accuracy of the developed methodology, vibrations were recorded at the Portuguese Railways Northern Line in a downward direction between the Porto (Campanhã) and Lisbon (Oriente) stations, a distance comprising a total of 275 km. Multiple AP and IC railway vehicles were monitored while performing regular passenger service. Then, the results were compared with those reported by the Portuguese infrastructure manager (IP) obtained by the EM 120 track inspection vehicle.

### 5.1. Measurement Campaigns

Floor measurements were taken on nine AP and six IC trains under normal passenger service conditions. Five measurements (three in AP and two in IC) were taken on each position inside the train: start, mid, and end cars/carriages, in seat locations near the rear bogies.

Figure 9 demonstrates both the AP and IC designs and the measurement locations. As discussed, the AP train was constituted of six cars. Thus, three acceleration measurements were taken at cars 1, 4, and 6, corresponding to the train's beginning, middle, and end cars. Moreover, car 1 was classified as comfort class, whereas cars 4 and 6 were standard classes. Thus, cars 4 and 6 presented an equal interior layout, unlike car 1. Figure 9a presents those interior layouts and the respective measurement locations. The IC train comprised five hauled Corail coaches; carriage 1 was classified as comfort class and carriages 3 and 5 were standard. These carriages corresponded to that type of train's beginning, middle, and end carriage, respectively. As on the AP train, the comfort and standard classes had different interior layouts, as presented in Figure 9b. Figure 9c shows the experimental setups on the AP (left side) and IC (right side) trains.

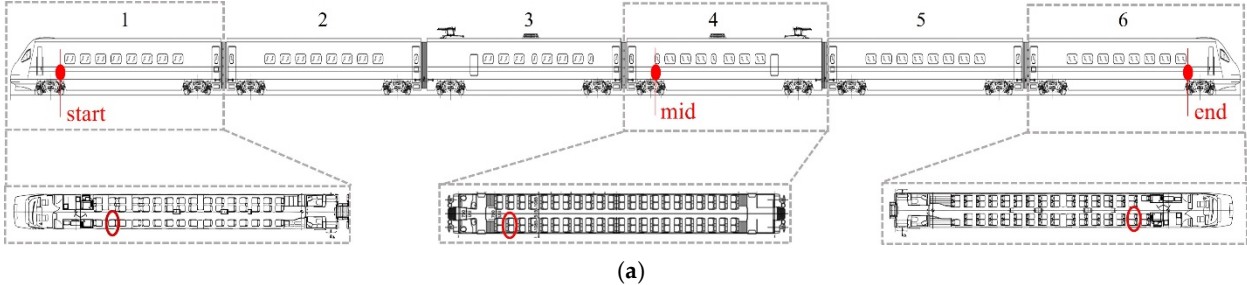

(a)

**Figure 9.** *Cont.*

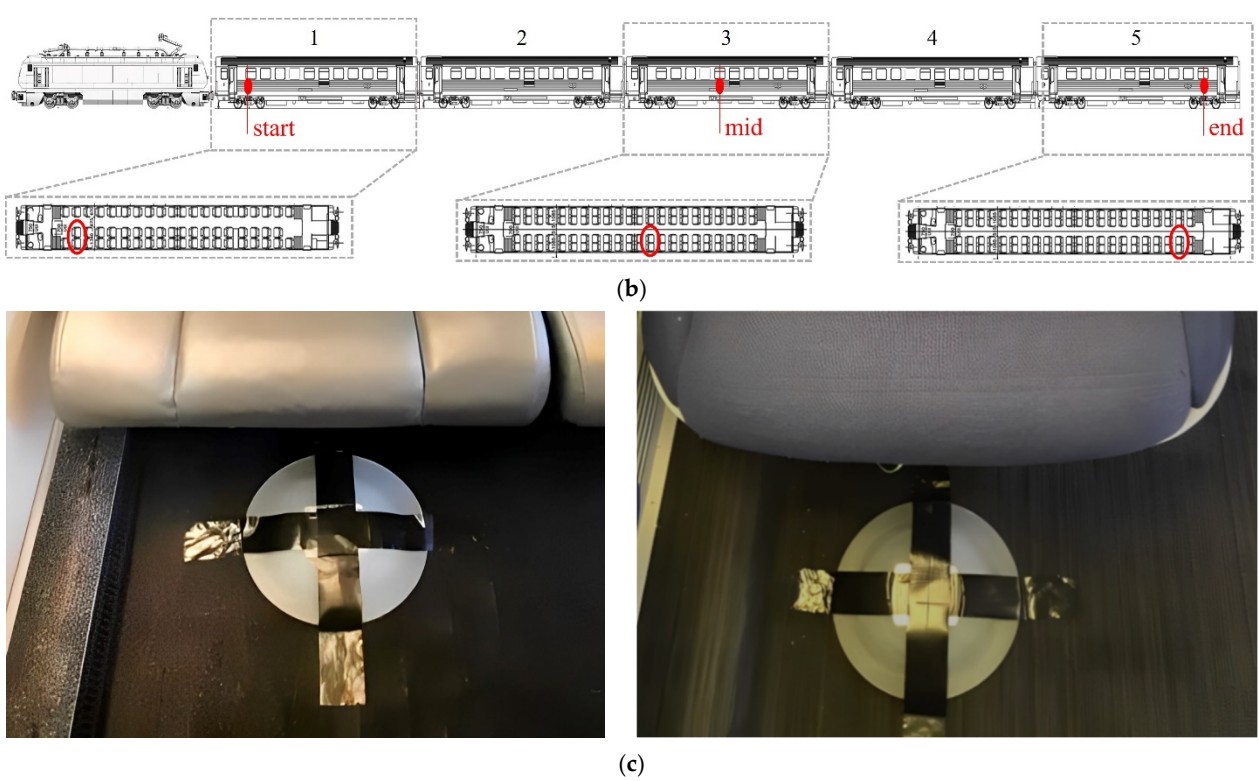

**(b)**

**(c)**

**Figure 9.** Floor measurement locations: (**a**) AP train and car measurement locations; (**b**) IC train and car measurement locations; (**c**) accelerometer placement on AP (left side) and IC (right side) trains.

### 5.2. Maintenance Needs Identified by IP

Currently, the Portuguese infrastructure manager IP assesses railway track infrastructure irregularities through measurements obtained by the passage of the EM 120 track inspection vehicle (Figure 10) [58,59]. This vehicle can detect and record track infrastructure faults with high precision.

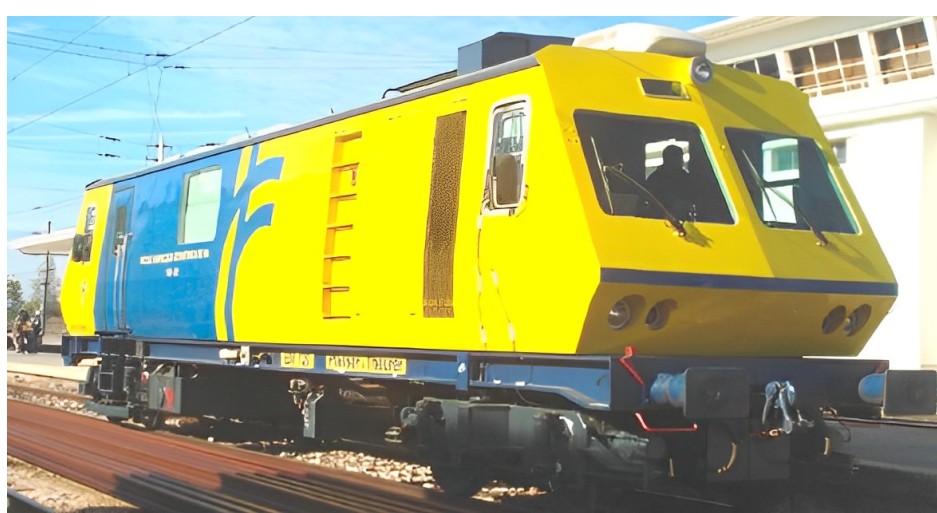

**Figure 10.** EM 120 track inspection vehicle from IP (adapted from [57]).

The most recently released report regarding the Portuguese Northern Line railway track infrastructure evaluation and maintenance needs detection by IP revealed multiple track sections in poor conditions and, consequently, with maintenance needs [59]. Aspects concerning those segments' identification can be found in Table 4.

**Table 4.** Track segments' maintenance needs identified by IP through the passage of EM 120 inspection vehicle (adapted from [56]).

| Track Section | | Kilometre Interval | | Track Segment |
|---|---|---|---|---|
| Start Station | End Station | Start | End | |
| Alhandra | Castanheira do Ribatejo | 26 | 27 | A |
| Albergaria dos Doze | Alfarelos | 147 | 199 | B |
| Pampilhosa | Válega | 232 | 297 | C |
| Válega | Espinho | 297 | 316 | D |

### 5.3. Maintenance Needs Identified by the Indirect Method

The data analysis identified railway track infrastructure maintenance sections by comparing the discomfort zones of the AP train versus those of the IC train. Karoumi et al. [37] and Norris [38] stated that a railway track infrastructure with isolated irregularities presents higher vibration levels at that location than a healthy one. Those higher vibrations will lead to increased discomfort levels.

The developed methodology divided floor discomfort levels into "Not uncomfortable" and "Uncomfortable" categories depending on the instantaneous acceleration evaluation. Those under 0.315 m/s$^2$, corresponding to the ISO 2631 first discomfort level threshold, were grouped in the first category, while the second category comprised those above that discomfort threshold. Therefore, an abnormality in the railway track infrastructure was expected to present discomfort peaks higher than 0.315 m/s$^2$ at that location. Conversely, a healthy infrastructure should have instantaneous floor discomfort levels lower than that threshold. Ideally, those discomfort levels should present consistent behaviour. The reported trend was observed in the acceleration records and analysis. Figure 11 illustrates AP and IC trains' behaviour regarding the same locations (latitude).

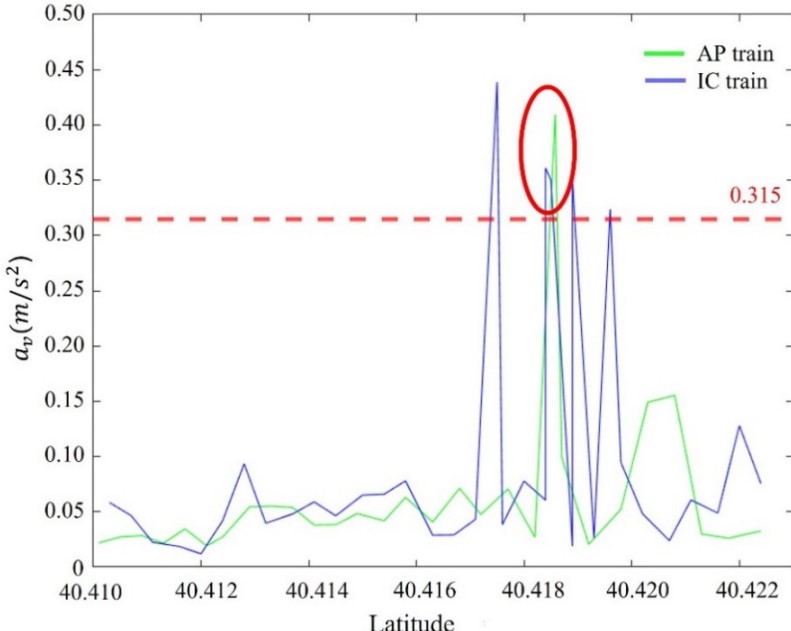

**Figure 11.** Instantaneous floor discomfort levels for AP and IC trains (where the red mark identifies a railway track infrastructure abnormality).

Vibration peaks capable of going through primary and secondary suspension systems cause passenger discomfort. AP and IC trains have different suspension systems and elements; thus, when the discomfort sections of both trains agree, it can be concluded that the railway track infrastructure requires maintenance. An example of that matching between both trains' discomfort levels is observed in Figure 11. The red mark illustrates

the overlapping between the AP and IC discomfort levels. Thus, it can be established that an abnormality is presented at that specific location, and the railway track infrastructure requires maintenance. Figure 11 shows all maintenance required sections identified on the railway Northern Line using this algorithm, where AP and IC floor discomfort levels matched. Those below a 220 m distance between peak locations corresponded to the sections where maintenance was needed and are identified and illustrated.

According to Figure 12, 12 maintenance zones were identified. For each zone, the start and end coordinates and their associated line kilometre are registered in Table 5, which also defines the matching with the IP-identified track sections. Due to the map resolution, some close zones appear to be just one. Blue and green markers define two sections that are, indeed, divided into small zones—1 and 2 for the blue marker and 3, 4, and 5 for the green section. The discomfort location obtained at the Porto (Campanhã) train station is not numbered, as it was due to the acceleration to start the motion and not to a railway infrastructure abnormality.

Ten listed locations match those that the Portuguese infrastructure manager company (IP) suggested to be subject to maintenance, identified by the EM 120 inspection vehicle [58]. Therefore, the obtained locations were critical points of the railway track infrastructure, thus validating this methodology. Moreover, two extra zones were detected: zones 10 and 11. In addition to its proven accuracy, the model demonstrated the ability to be considered a viable alternative to current methods. It should be highlighted that it is a low-cost method capable of simultaneously identifying railway track infrastructure isolated irregularities and analysing passengers' comfort levels.

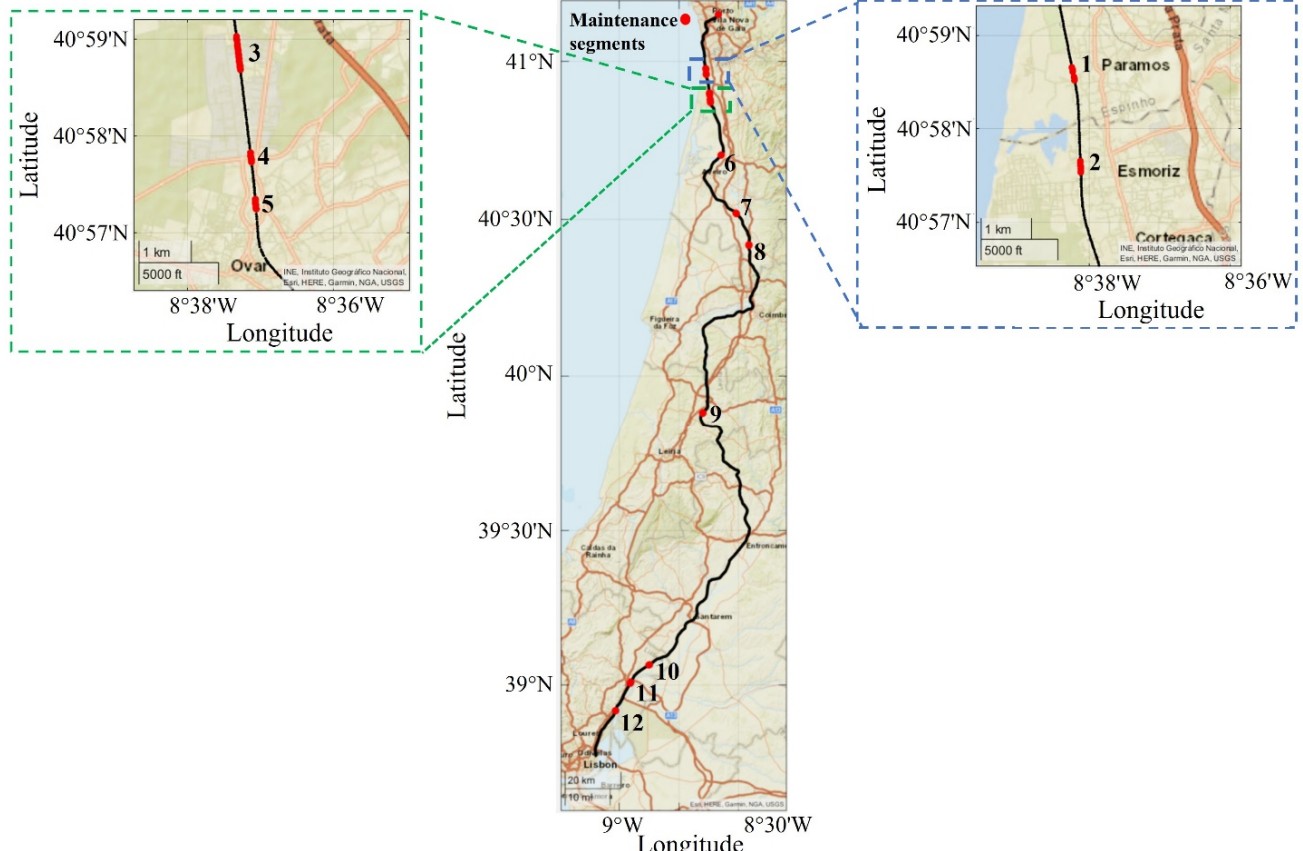

**Figure 12.** Nothern Line railway track infrastructure maintenance needs identification; detailed zones (left side and right side) are at the laterals.

**Table 5.** Maintenance needs identification segments and comparison with EM 120 inspection vehicle results.

| Zone Number | Initial Coordinates | | Final Coordinates | | Line km | Track Segment IP |
|---|---|---|---|---|---|---|
| | Latitude | Longitude | Latitude | Longitude | | |
| 1 | 40.9776 | −8.6374 | 40.9753 | −8.6368 | 315 | D |
| 2 | 40.9609 | −8.6354 | 40.9589 | −8.6353 | 313 | D |
| 3 | 40.9005 | −8.6221 | 40.8948 | −8.6212 | 306 | D |
| 4 | 40.8805 | −8.6189 | 40.8789 | −8.6187 | 304 | D |
| 5 | 40.8725 | −8.6178 | 40.8708 | −8.6176 | 303 | D |
| 6 | 40.7044 | −8.5727 | 40.7038 | −8.5735 | 283 | C |
| 7 | 40.5196 | −8.5108 | 40.5180 | −8.5087 | 255 | C |
| 8 | 40.4198 | −8.4564 | 40.4172 | −8.4566 | 242 | C |
| 9 | 39.8820 | −8.6489 | 39.8807 | −8.6507 | 166 | B |
| 10 | 39.0651 | −8.8734 | 39.0639 | −8.8758 | 46 | - |
| 11 | 39.0081 | −8.9518 | 39.0047 | −8.9542 | 37 | - |
| 12 | 38.9159 | −9.0155 | 38.9149 | −9.0160 | 26 | A |

It should be noted that the proposed methodology cannot characterise the abnormality types present on railway track infrastructure, but is focused on detecting critical track sections. Additionally, the present study did not consider the effect of aerodynamic forces, which has a more significant effect on high-speed trains, namely, those achieving speeds faster than 300 km/h, which is not the case in this experimental research [60]. Nevertheless, it should be highlighted that since AP and IC trains have different structural designs, their aerodynamic behaviours are different, which may have a distinct influence on their vibration levels [61].

In the future, more work is intended to be developed regarding abnormality identification using more advanced methodologies, such as supervised machine learning procedures. Implementing these procedures will require vehicles to pass through track sections with well-known and characterized abnormalities. This way, it will be possible to learn and identify the typical dynamic response patterns of each abnormality [62].

## 6. Conclusions

Railways are currently one of the most used mass transportation systems worldwide. The increase in the use of railway transportation demands more velocity and load capacity for trains. These two factors and weather conditions accelerate railway track infrastructure degradation, which must be assessed more frequently.

The interaction between rail infrastructure, wheels, and vehicle motion creates a complex vibration environment, which is filtered by two suspension systems: primary and secondary. The former intends to induce stability and reduce the vibration transmission from track–wheels interactions, while the latter concerns the bogie–carbody transmission. These systems decrease the vibration transmission to the passenger and thus increase comfort.

Railway track infrastructure, the main component of the rail industry, affects both safety and comfort; thus, it requires continuous maintenance. Currently, railway track infrastructure evaluation is mainly accomplished by inspection vehicles. However, these vehicles are expensive, and their passage disrupts programmed timetables. Based on these limitations, a low-cost CBM system was developed to identify railway track infrastructure maintenance needs based on comfort measurements without timetable disruptions.

AP and IC trains ran multiple journeys along the Portuguese Northern Line, where accelerations and GPS measurements were recorded. ISO 2631 standard methodology was followed to obtain floor discomfort levels. Matching the instantaneous floor discomfort for both types of trains, railway track infrastructure maintenance sections were found. The identified geographic locations were similar to those obtained by the EM 120 inspection vehicle. Therefore, the system was validated and proven to be precise. Moreover, compared

with previous studies where acceleration measurements were acquired at the axle box, the application of comfort levels to assess maintenance requirements is a novelty.

The developed system provides a complementary, low-cost, CBM railway track infrastructure analysis capable of detecting abnormalities (although not the type of) by using in-service passenger trains to obtain measurements, thus avoiding any service disruption. A future goal of the present research is to identify the abnormality type. Therefore, machine learning procedures will be developed and applied.

**Author Contributions:** Conceptualization, P.S., P.P. and J.M.; methodology, P.S. and P.P.; software, P.S. and P.P.; validation, P.S., P.P., D.R., J.M. and E.S.; formal analysis, P.S.; investigation, P.S.; resources, P.S.; data curation, P.S.; writing—original draft preparation, P.S.; writing—review and editing, P.P., D.R., J.M. and E.S.; visualization, P.S.; supervision, J.M. and E.S.; project administration, J.M.; funding acquisition, P.S. and D.R. All authors have read and agreed to the published version of the manuscript.

**Funding:** This research was funded by Fundação para a Ciência e Tecnologia, grant number PD/BD/143161/2019. The authors also acknowledge the financial support from the Base Funding-UIDB/04708/2020 and Programmatic Funding-UIDP/04708/2020 of the CONSTRUCT—Instituto de Estruturas e Construções, funded by national funds through the FCT/MCTES (PIDDAC).

**Institutional Review Board Statement:** Not applicable.

**Informed Consent Statement:** Not applicable.

**Data Availability Statement:** Data is unavailable due to privacy restrictions.

**Acknowledgments:** This work is a result of the project "FERROVIA 4.0", reference POCI-01-0247-FEDER- 046111, co-funded by the European Regional Development Fund (ERDF) through the Operational Programme for Competitiveness and Internationalization (COMPETE 2020) and the Lisbon Regional Operational Programme (LISBOA 2020) under the PORTUGAL 2020 Partnership Agreement. The first author thanks Fundação para a Ciência e Tecnologia (FCT) for a PhD scholarship under the project iRail (PD/BD/143161/2019). The authors would like to acknowledge the support of the projects FCT LAETA–UIDB/50022/2020, UIDP/50022/2020, and UIDB/04077/2020. No potential competing interest was reported by the authors.

**Conflicts of Interest:** The authors declare no conflict of interest.

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
