# Peer review of "Indirect Assessment of Railway Infrastructure Anomalies Based on Passenger Comfort Criteria"

_applsci, doi:10.3390/app13106150_

Round 1

Reviewer 1 Report

In this study, a methodology capable of detecting railway track infrastructure failures is proposed based on the discomfort level. The paper is interesting. Several minor issues are:

1. The title of paper is not suitable. It needs to be revised.

2. The aerodynamics may affect the vibration. Therefore, the ignorance of the train aerodynamics is one of the weaknesses of the paper. The minor issue can be discussed in the Conclusions. Two references are advised to add in this paper. â‘ influence of marshalling length on aerodynamic characteristics of urban EMUs under crosswind, JAFM; â‘¡ Numerical Study on Aerodynamic Resistance Reduction of High-speed Train Using Vortex Generator, EACFM. 

Author Response

The authors acknowledge the positive evaluation made by the Reviewer. Concerning the pertinent remarks raised by the Reviewer (italic), the following changes were introduced into the manuscript, or the following clarification comments are made:

“In this study, a methodology capable of detecting railway track infrastructure failures is proposed based on the discomfort level. The paper is interesting. Several minor issues are:”

  • The title of paper is not suitable. It needs to be revised.

The authors acknowledge the observation addressed by the Reviewer.

Based on the Reviewer suggestion, the manuscript title was revised, and the following was suggested:

“Indirect assessment of railway infrastructure anomalies based on passenger comfort criteria”

  • The aerodynamics may affect the vibration. Therefore, the ignorance of the train aerodynamics is one of the weaknesses of the paper. The minor issue can be discussed in the Conclusions. Two references are advised to add in this paper. â‘ influence of marshalling length on aerodynamic characteristics of urban EMUs under crosswind, JAFM; â‘¡ Numerical Study on Aerodynamic Resistance Reduction of High-speed Train Using Vortex Generator, EACFM.

The authors acknowledge the observation addressed by the Reviewer and updated the manuscript by introducing the aerodynamical influence on vibration levels. The following was introduced:

“5.3 Maintenance needs identified by the indirect method

comfort levels. Additionally, the present study does not consider the effect of aerodynamic forces. That has a more significant effect on high-speed trains, namely those achieving speeds higher than 300 km/h, which is not the case in this experimental research [58]. Nevertheless, it should be highlighted that since AP and IC trains have different structural designs, their aerodynamic behaviours are different, which may have a distinct influence on their vibration levels [59].

References

  1. Li T, Liang H, Zhang J, Zhang J. Numerical study on aerodynamic resistance reduction of high-speed train using vortex generator. Engineering Applications of Computational Fluid Mechanics 2023;17. https://doi.org/10.1080/19942060.2022.2153925.

  1. Liang H, Sun Y, Li T, Zhang J. Influence of Marshalling Length on Aerodynamic Characteristics of Urban Emus under Crosswind. Journal of Applied Fluid Mechanics 2023;16. https://doi.org/10.47176/jafm.16.01.1338.

Thanks for your time and consideration.

Sincerely,

Patricia Silva

(Corresponding author)

CONSTRUCT-LESE

Faculty of Engineering – University of Porto

Rua Dr. António Bernardino de Almeida, 431

4249-015 Porto, Portugal

e-mail:[email protected]

Reviewer 2 Report

1. The research content is too simple and does not provide a clear explanation for why the vehicle vibration performance exceeds the standard. there could be various causes leading to this issue such as rail defects, wheel defects, or vehicle configuration factors. Thus, it is essential to determine conclusively that rail abnormalities caused the observed deviation.

2. Poor novelty in the content. Similar research can be found which basing on the characteristics of axle box vibrations and analyzed the type of track faults caused by these vibrations. This work does not give new information.

3. The validation of conclusions lacks support from field results. please provide an explanation of the actual situations of track faults in the relevant section.

Author Response

The authors acknowledge the positive evaluation made by the Reviewer. Concerning the pertinent remarks raised by the Reviewer (italic), the following changes were introduced into the manuscript, or the following clarification comments are made:

  • The research content is too simple and does not provide a clear explanation for why the vehicle vibration performance exceeds the standard. there could be various causes leading to this issue such as rail defects, wheel defects, or vehicle configuration factors. Thus, it is essential to determine conclusively that rail abnormalities caused the observed deviation.

The authors acknowledge the observation addressed by the Reviewer.

The authors would like to emphasise that the main objective of this work was to develop a methodology capable of detecting railway infrastructure abnormalities/damage based on passengers’ comfort levels.

Although simple, this methodology was never applied to assess maintenance requirements in a railway environment, representing a novelty regarding current bibliographic research.

Furthermore, the results demonstrated a high correlation between the discomfort levels and the track sections identified by the infrastructure manager as requiring maintenance. Also, the results show the methodology’s potential to be used for infrastructure condition assessment based on in-service trains.

Currently, this study doesn’t intend to determine the causes for the abnormalities/damages responsible for the maintenance needs.

This step forward will require using more advanced methodologies, namely the ones based on supervised machine learning procedures. Implementing these procedures will require the passage of the vehicles on track sections with known and well-characterised abnormalities, in order to learn the dynamic response patterns typical of each one of the anomalies.

Based on the Reviewer suggestion, the manuscript was updated with the following:

"5.3 Maintenance needs identified by the indirect method

It should be noted that the proposed methodology cannot characterise the abnormality types present on the railway track infrastructure, but is focused on detecting the critical track sections.

In the future, more work is intended to be developed regarding abnormality identification using more advanced methodologies, such as supervised machine learning procedures. Implementing these procedures will require vehicles to pass through track sections with well-known and characterised abnormalities. This way, it will be possible to learn and identify the typical dynamic response patterns of each abnormality [60].

References

  1. Meixedo A, Santos J, Ribeiro D, Calçada R, Todd MD. Online unsupervised detection of structural changes using train–induced dynamic responses. Mech Syst Signal Process 2022;165:108268. https://doi.org/10.1016/j.ymssp.2021.108268.

  1. Conclusions

Therefore, the system was validated and proved its precision. Moreover, compared with previous studies where acceleration measurements were acquired at the axle box, the application of comfort levels to assess maintenance requirements is a novelty.

A future goal of the present research is to identify the abnormality type. Therefore, machine learning procedures will be developed and applied.

"

  • Poor novelty in the content. Similar research can be found which basing on the characteristics of axle box vibrations and analyzed the type of track faults caused by these vibrations. This work does not give new information.

The authors acknowledge the observation addressed by the Reviewer.

The article proposes a novel methodology based on the vibrations measured at the carbody to assess possible infrastructure maintenance needs. This proposal is a clear step forward in relation to the existent studies that are practically all based on the axle box vibrations, as stated by the Reviewer.

The novelties and contributions of the present study are the following:

- Development of methodology capable of detecting rail track infrastructure abnormalities or damages based on measurements on in-service trains. This way, the service runs under regular operation without disruption or interference.

- Development of an easy-to-adapt and implement methodology, capable of being applied in any in-service railway vehicle. This way, overcoming the limitations of the accelerometer installation in the axle box, which imposes difficulties on the processes of installation, maintenance and dismounting.

- The results of the present methodology are not dependent on the vehicle type. Thus, it can be applied to passenger trains with different characteristics. This was stated based on the results of a large extent of measurement campaigns proving the methodology's accuracy and precision. 

Following the revisor suggestions, the following was updated:

"1. Introduction

More recently, cargo vehicles are also being used as instrumented inspection vehicles, particularly in the axle box [17–21]. However, implementing the experimental setup is problematic for that approach, limiting its use in passenger trains.  

Up to this date, passenger comfort levels have not been used for possible damage detection. Based on the close…

This study considers that the railway track infrastructure requires maintenance actions if multiple trains with different dynamic characteristics present floor discomfort on the same GPS location.

Based on the limitations of the previous experiments, this study aims to give clear contributions about some aspects that presently, according to the authors' knowledge, are not sufficiently addressed in the existing literature, particularly:

- Development of methodology capable of detecting rail track infrastructure abnormalities or damages based on measurements on in-service trains. This way, the service runs under regular operation without disruption or interference.

- Development of an easy-to-adapt and implement methodology, capable of being applied in any in-service rail vehicle. This way, overcoming the limitations of the accelerometer installation in the axle box, which imposes difficulties on the processes of installation, maintenance and dismounting.

- The results of the present methodology are not dependent on the vehicle type. Thus, it can be applied to passenger trains with different characteristics. This was stated based on the results of a large extent of measurement campaigns proving the methodology's accuracy and precision. 

Performing the Porto (Campanhã) – Lisbon (Oriente) connection on the …”

  • The validation of conclusions lacks support from field results. please provide an explanation of the actual situations of track faults in the relevant section.

The authors acknowledge the observation addressed by the Reviewer.

That is quite a pertinent question, especially when a future goal is to identify the abnormality type. However, information on this topic is confidential and not shared by the infrastructure manager. The manager only shares the sections needing maintenance, not their origin. This way, the presently obtained results were matched with those identified by the Infraestruturas de Portugal when passing the EM-120 inspection vehicle. Furthermore, future work to identify the abnormalities type will require passing rail vehicles through sections with previously identified and characterised abnormalities. Their dynamic behaviour will support the learning procedure.

Considering the reviewer suggests the manuscript was updated as follows:

“5.3 Maintenance needs identified by the indirect method

In the future, more work is intended to be developed regarding abnormality identification using more advanced methodologies, such as supervised machine learning procedures. Implementing these procedures will require vehicles to pass through track sections with well-known and characterised abnormalities. This way, it will be possible to learn and identify the typical dynamic response patterns of each abnormality [60].

  1. Conclusions

any service disruption. A future goal of the present research is to identify the abnormality type. Therefore, machine learning procedures will be developed and applied.

Thanks for your time and consideration.

Sincerely,

Patricia Silva

(Corresponding author)

CONSTRUCT-LESE

Faculty of Engineering – University of Porto

Rua Dr. António Bernardino de Almeida, 431

4249-015 Porto, Portugal

Reviewer 3 Report

1)It is suggested to remove the sentence ‘… up to 850 km’ in the first paragraph of the introduction. In some large-scale high-speed networks, like China, it is very common to schedule a rail service over 1000 km or even 2000 km.

2)Apart from the purpose of improving comfort, the maintenance of the track can also improve the interaction performance of vehicle and overhead infrastructures as reported in [*], which is also important to enable a safe and stable rail service. It is recommended to indicate this to give the readers a full picture of this problem though this is not the focus of this work.

[*] Song, Yang, et al. "A spatial coupling model to study dynamic performance of pantograph-catenary with vehicle-track excitation." Mechanical Systems and Signal Processing 151 (2021): 107336.

3) According to [&], the vibration at a certain frequency which is close to the human body's natural frequency has the dominant effect in affecting comfort. Could you please comment on how to consider the frequency issue in this work when implementing the assessment?

[&] Demić, M., J. Lukić, and Ž. Milić. "Some aspects of the investigation of random vibration influence on ride comfort." Journal of sound and vibration 253.1 (2002): 109-128.

4) Is there any difference to do the assessment using the present method when dealing with ballasted and ballastless tracks?

5) It is noticed that 0.315 m/s^2 is the first level of comfort threshold. What are the other levels? A brief introduction may be helpful for readers to follow.

Author Response

The authors acknowledge the positive evaluation made by the Reviewer. Concerning the pertinent remarks raised by the Reviewer (italic), the following changes were introduced into the manuscript, or the following clarification comments are made:

  • It is suggested to remove the sentence ‘… up to 850 km’ in the first paragraph of the introduction. In some large-scale high-speed networks, like China, it is very common to schedule a rail service over 1000 km or even 2000 km.

The authors acknowledge the observation addressed by the Reviewer and removed the sentence, as follows:

“1. Introduction

Additionally, due to its low environmental impact, multiple governments are promoting its use as a mass transportation mode, especially for connecting cities with distances up to 850 km. Following… “

  • Apart from the purpose of improving comfort, the maintenance of the track can also improve the interaction performance of vehicle and overhead infrastructures as reported in [*], which is also important to enable a safe and stable rail service. It is recommended to indicate this to give the readers a full picture of this problem though this is not the focus of this work.

[*] Song, Yang, et al. "A spatial coupling model to study dynamic performance of pantograph-catenary with vehicle-track excitation." Mechanical Systems and Signal Processing 151 (2021): 107336.  

The authors acknowledge the observation addressed by the Reviewer and updated the manuscript by introducing the improvement of the overhead infrastructures achieved by track maintenance. The following was introduced:

“1. Introduction

reduced cost. Moreover, railway track infrastructure maintenance also improves the interaction performance of the rail vehicle and the overhead infrastructures, such as the pantograph-catenary interaction, which leads to a safer, more stable and comfortable journey [22].

References

  1. Song Y, Wang Z, Liu Z, Wang R. A spatial coupling model to study dynamic performance of pantograph-catenary with vehicle-track excitation. Mech Syst Signal Process 2021;151:107336. https://doi.org/10.1016/j.ymssp.2020.107336.

  • According to [&], the vibration at a certain frequency which is close to the human body's natural frequency has the dominant effect in affecting comfort. Could you please comment on how to consider the frequency issue in this work when implementing the assessment?

[&] Demić, M., J. Lukić, and Ž. Milić. "Some aspects of the investigation of random vibration influence on ride comfort." Journal of sound and vibration 253.1 (2002): 109-128.

The authors acknowledge the observation addressed by the Reviewer.

Indeed the human body has its natural vibration frequency, which depends on the body region. If an externally induced vibration matches that body natural frequency resonance may occur, which may lead not only to discomfort but also to tissue damage. In the present work, the standard ISO 2631 was followed. That considers the application of Wk weighting curve for floor measurements, which defines the frequencies within 4-12 Hz as the more sensitive ones.

In the present study, a maintenance section was only identified if multiple Alfa Pendular and Intercity trains presented floor discomfort levels at that location. Therefore, due to the number of experiments performed (15) it is quite unlike that all journeys noticed floor discomfort at the same exact location unless there is an abnormality in the railway track infrastructure.

Following the revisor recommendations, the article was updated as follows:

“4.1 Whole-body vibration evaluation

Weighting curves application relies on the measurement location and purpose. This way, the effect of the frequencies most influencing passengers' discomfort is amplified, while that of the frequencies less impacting discomfort is reduced. The total… “

  • Is there any difference to do the assessment using the present method when dealing with ballasted and ballastless tracks?

The authors acknowledge the observation addressed by the Reviewer.

The ISO 2631 calculations, which were the basis of the present study, are not defined based on the track type. Thus, there are no expected assessment differences when applying the present method in ballastless tracks. The proposed methodology was only employed in ballasted tracks once the Portuguese railway line is of this type.

The manuscript was updated by introducing the following:

“4. Indirect method for infrastructure condition assessment based on comfort criteria

needs locations. It should be highlighted that the methodology was applied on ballasted tracks once, that is, the track type of the Northern line of the Portuguese railways. Nevertheless, there are no expected assessment differences when applying the present method into ballastless tracks.”

  • It is noticed that 0.315 m/s^2 is the first level of comfort threshold. What are the other levels? A brief introduction may be helpful for readers to follow.

The authors acknowledge the observation addressed by the Reviewer.

The other comfort levels defined by the standard are presented in section “4.1 Whole body vibration evaluation”, specifically in Table 3. Nevertheless, a reference to that section was introduced, namely:

“4.1 Whole body vibration measurements

Lastly, based on , a defined scale evaluates the discomfort levels (see Table 3) in which higher than 0.315 m/s2 values are considered uncomfortable.

Table 3. - ISO 2631 comfort evaluation scale. Adapted from [46].

Ride comfort

≤ 0.315

Not uncomfortable

0.5 – 0.63

Little uncomfortable

0.63 – 0.8

Little uncomfortable to fairly uncomfortable

0.8 – 1.0

Fairly uncomfortable to uncomfortable

1.0 – 1.25

Uncomfortable

1.25 – 1.6

Uncomfortable to very uncomfortable

1.6 – 2.0

Very uncomfortable

2.0 – 2.5

Very uncomfortable to extremely uncomfortable

4.2.2 Data analysis

Those were defined based on the first discomfort threshold of ISO 2631 standard. Table 3 presents all comfort levels defined according to the standard. The first category … “

Thanks for your time and consideration.

Sincerely,

Patricia Silva

(Corresponding author)

CONSTRUCT-LESE

Faculty of Engineering – University of Porto

Rua Dr. António Bernardino de Almeida, 431

4249-015 Porto, Portugal

Round 2

Reviewer 2 Report

The addressed issues have been responded and improvements can be seen now. Regarding the test method on-track for a railway vehicle, the same topic in the journal Vehicle System Dynamics was seen with the test method and ride comfort evaluation presented.  It is suggested to have more details about the data acquisition and data processing be presented.

Shi H, Wang J, Wu P, et al. Field measurements of the evolution of wheel wear and vehicle dynamics for high-speed trains. Vehicle System Dynamics, 2018, 56(8), 1187-1206.

Author Response

The authors acknowledge the positive evaluation made by the Reviewer. Concerning the pertinent remarks raised by the Reviewer (italic), the following changes were introduced into the manuscript, or the following clarification comments are made:

  • The addressed issues have been responded and improvements can be seen now. Regarding the test method on-track for a railway vehicle, the same topic in the journal Vehicle System Dynamics was seen with the test method and ride comfort evaluation presented. It is suggested to have more details about the data acquisition and data processing be presented.

Shi H, Wang J, Wu P, et al. Field measurements of the evolution of wheel wear and vehicle dynamics for high-speed trains. Vehicle System Dynamics, 2018, 56(8), 1187-1206.

The authors acknowledge the observation addressed by the Reviewer.

Based on the Reviewer suggestion, the manuscript was updated with the following:

"4.2.1 Data acquisition system

card. Sensors were mounted on board of multiple trains. Its location inside the rail train was defined based on points corresponding to the train's beginning, middle and end. Moreover, each of these locations was defined rear the respective bogie.

Floor vibration measurements were conducted by the three-axial accelerometer (PCE-VDL-24I), with a measurement range of ±16g, resolution of 0.004g and sampling rate between 0 – 2400 Hz, which permits data recording and follows ISO 2631 requirements [52].

For future data processing analysis, it is fundamental to synchronise data; thus, vibration and geographic measurements occur synchronously. Figure 6

As aforementioned, experiments run under multiple AP and IC trains, which have different structural designs. Figure 7 illustrates both types of trains.

(a)

(b)

Figure 7. – Experimented vehicles: (a) AP train; (b) IC train.

4.2.2. Data analysis

The algorithm starts by calculating the instantaneous floor discomfort (discomfort level at each second) according to ISO 2631 recommendations for each rail journey. A slight adaptation of the standard was conducted, as discomfort levels were divided only into two categories instead of the nine defined for the standard. Those categories were defined based on the first discomfort threshold of the ISO 2631 standard.

This way, geographic location and acceleration records are acquired at the same location. However,

5.Case study: Portuguese Northern Line condition assessment

To validate and verify the accuracy of the developed methodology, vibration records took place at the Portuguese Railways Northern Line, downward direction, connecting Porto (Campanhã) to Lisbon (Oriente) stations and comprising a total of 275 km.

5.1. Measurements campaigns

Five measurements (three in AP and two in IC) were taken on each position inside the train: start, mid and end cars/carriages, specifically in seat locations near the rear bogies.

References

  1. Shi H, Wang J, Wu P, Song C, Teng W. Field measurements of the evolution of wheel wear and vehicle dynamics for high-speed trains. Vehicle System Dynamics 2018;56:1187–206. https://doi.org/10.1080/00423114.2017.1406963.

"

Thanks for your time and consideration.

Sincerely,

Patricia Silva

(Corresponding author)

CONSTRUCT-LESE

Faculty of Engineering – University of Porto

Rua Dr. António Bernardino de Almeida, 431

4249-015 Porto, Portugal

Reviewer 3 Report

Thanks for good revision.

Author Response

The authors acknowledge the positive evaluation made by the Reviewer. Concerning the pertinent remarks raised by the Reviewer (italic), the following changes were introduced into the manuscript, or the following clarification comments are made:

  • Thanks for good revision.

The authors acknowledge the positive observation addressed by the Reviewer.

Thanks for your time and consideration.

Sincerely,

Patricia Silva

(Corresponding author)

CONSTRUCT-LESE

Faculty of Engineering – University of Porto

Rua Dr. António Bernardino de Almeida, 431

4249-015 Porto, Portugal
